ecology

citizen science, biodiversity, spatial and temporal sampling, dynamic models, predictive modelling

**Author for correspondence:**
Corey T. Callaghan
e-mail: c.callaghan@unsw.edu.au

# Optimizing future biodiversity sampling by citizen scientists

Corey T. Callaghan[1,2,3], Alistair G. B. Poore[2], Richard E. Major[1,3], Jodi J. L. Rowley[1,3] and William K. Cornwell[1,2]

[1]Centre for Ecosystem Science, School of Biological, Earth and Environmental Sciences, and [2]Ecology & Evolution Research Centre, School of Biological, Earth and Environmental Sciences, UNSW Sydney, Sydney, New South Wales 2052, Australia
[3]Australian Museum Research Institute, Australian Museum, Sydney, New South Wales 2010, Australia

CTC, 0000-0003-0415-2709

We are currently in the midst of Earth's sixth extinction event, and measuring biodiversity trends in space and time is essential for prioritizing limited resources for conservation. At the same time, the scope of the necessary biodiversity monitoring is overwhelming funding for professional scientific monitoring. In response, scientists are increasingly using citizen science data to monitor biodiversity. But citizen science data are 'noisy', with redundancies and gaps arising from unstructured human behaviours in space and time. We ask whether the information content of these data can be maximized for the express purpose of trend estimation. We develop and execute a novel framework which assigns every citizen science sampling event a marginal value, derived from the importance of an observation to our understanding of overall population trends. We then make this framework predictive, estimating the expected marginal value of future biodiversity observations. We find that past observations are useful in forecasting where high-value observations will occur in the future. Interestingly, we find high value in both 'hotspots', which are frequently sampled locations, and 'coldspots', which are areas far from recent sampling, suggesting that an optimal sampling regime balances 'hotspot' sampling with a spread across the landscape.

## 1. Introduction

Assessing biodiversity trends in space and time is essential for conservation [1–5]. Reliable biodiversity trend estimates, at multiple spatial scales [6], allow us to track our global progress in curbing biodiversity loss while managing our scarce conservation resources [1]. Unsurprisingly, reliable trend estimates are best derived from long-term [7,8], well-designed surveys, carried out over a wide spatial and temporal scale [1,9,10]. But scientific funding for long-term ecological and conservation research is failing to keep pace with conservation needs [11,12]. Increasingly, government agencies, scientific researchers and conservationists are turning to citizen science data to help inform the state of biodiversity at local [13–16], regional [17,18] and global scales [19–21].

Citizen science—the cooperation between scientific experts and non-experts—is an incredibly diverse and rapidly expanding field [22–24]. Projects generally fall along a continuum based on the level of associated structure [25,26], ranging from unstructured (e.g. opportunistic or incidental projects which require little to no training; iNaturalist) to structured (e.g. projects with specific objectives, rigorous protocols and survey design; UK Butterfly Monitoring Scheme). The level of structure, in turn, influences the degree of redundancies and gaps in the data, as well as the overall data quality of a particular project. For instance, observer skill [27], number of participants in a group and the technological capabilities of a participant may influence the

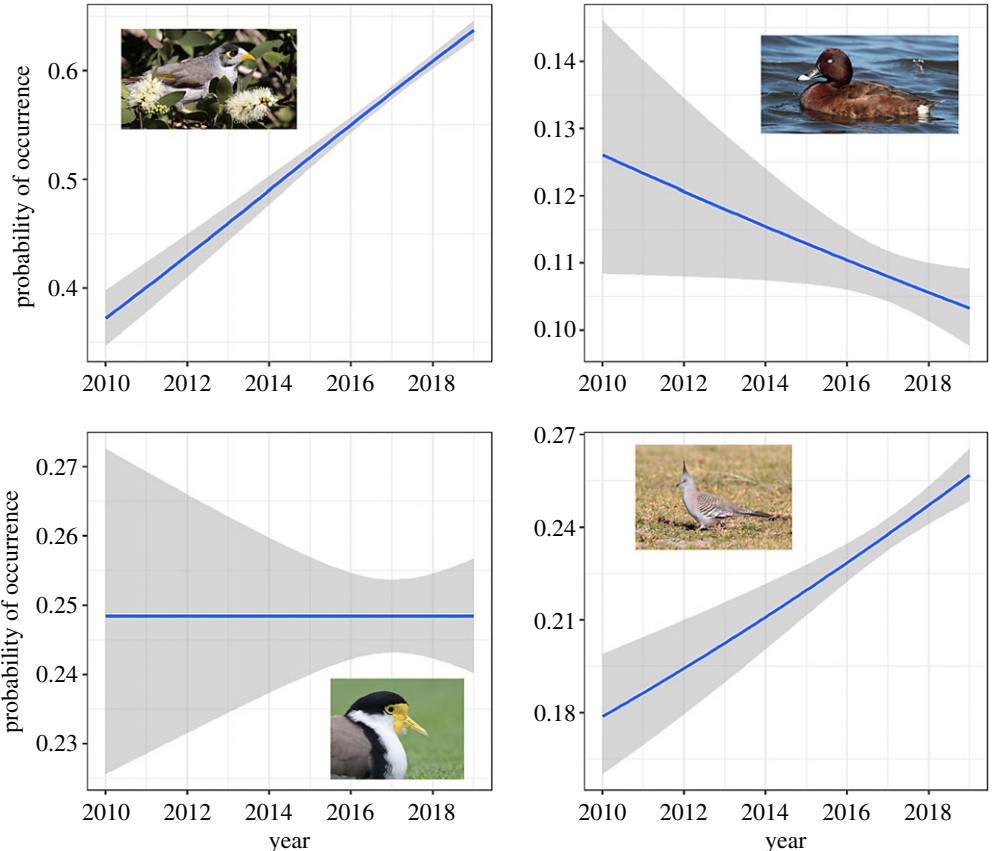

**Figure 1.** The ultimate goal in understanding population trends is to minimize the uncertainty for a population trend model, providing more robust measures of population trends. Shown here are four example population trend models, based on eBird data between 2010 and 2018, for noisy miner (top left), hardhead (top right), masked lapwing (bottom left) and crested pigeon (bottom right), in the Greater Sydney Region, Australia. Each model incorporates approximately 26 000 biodiversity sampling events. (Online version in colour.)

data collected by some, but not necessarily all, citizen science projects. Generalizable among citizen science projects, however, are various redundancies and gaps (i.e. spatial and temporal biases) [28,29]. Observers submitting observations on weekends [30], sampling near roads and human settlements [31], and visiting known 'hotspots' for biodiversity [32] are all examples of how unstructured human behaviour leads to redundancies and gaps in citizen science data [33]. These biases are not restricted to citizen science projects. Indeed, our historical understanding of biodiversity is also biased due to variation in sampling effort, evident from natural history collections [28,34]. Many sampling methods have been proposed to optimize biodiversity sampling by professional scientists [35–39], frequently dependent on spatial scale [40]. While structured citizen science projects often adapt some aspects of optimal sampling in their methods (e.g. stratified sampling), little attention has been given to optimal sampling in unstructured citizen science projects [1].

One of the reasons that optimal sampling has been largely ignored in unstructured citizen science projects is because redundancies and gaps in the data are seen as a 'necessary hurdle' [41]. Also, in the case of broad-scale biodiversity data collected at large voluminous scales, the biases can generally be accounted for statistically [42,43]; for instance, by filtering or subsetting data [44], pooling multiple data sources [45], or machine learning and hierarchical clustering techniques [31,46]. Indeed, despite known biases, citizen science data have increased our knowledge of species distribution models [47,48], niche breadth [49], biodiversity [20,50],

phenological research [51,52], invasive species detection [53,54] and phylogeographical research [55,56].

Despite the potential outcomes from citizen science data, estimating biodiversity trends is perhaps the most important, given the current need to efficiently and effectively monitor biodiversity [1,4,5]. From a conservation perspective, the goal is relatively straightforward: provide robust measures of species' trends through time, a critical component of the IUCN Red List index [57]. Estimating trends with citizen science data is best done with data from structured projects (i.e. less biases to account for, generally resulting in greater certainty) [18]. But unstructured and semistructured projects are increasingly harnessed for trend detection [10,58–62]. The robustness of these trend estimates is critical, and the goal should be to continuously decrease the uncertainty surrounding these estimates (e.g. figure 1). Unsurprisingly, uncertainty is generally related to the number of observations, as well as appropriate sampling, through time (e.g. https://github.com/coreytcallaghan/optimizing-citizen-science-sampling/blob/master/Figures/Noisy_miner_gif.gif).

The number of citizen science projects which are focused on ecological and environmental monitoring is increasing [14,24], highlighting the potential that citizen science holds for the future of ecology, conservation and natural resource management [20,63–65]. But a major obstacle in the future use of citizen science data remains understanding how to best extract information from 'noisy' citizen science datasets [41]. As mentioned, this noise from citizen science [29] can sometimes be alleviated using 'big data' statistical approaches [31], but this is most applicable for data originating

from large, successful citizen science projects—with lots of data. Even with big data, this is not always possible. But what about projects that are just beginning? Or projects focusing on taxa that are less popular with the general public [66,67]? Are there optimal strategies for sampling in space and time for estimating biodiversity trends?

Here, we investigate these questions with a specific objective: assess how spatial and temporal sampling by citizen scientists influences trend detection of biodiversity. Our approach is dynamic: we are interested in the parameters that influence the value of a given citizen science sampling event in both time and space. To do this, we (i) used 25 995 eBird citizen science sampling events, (ii) analysed linear trends for 235 species, (iii) calculated a measure of statistical leverage (i.e. marginal value)—the influence a given observation has on the population trend model of a species—for all checklists for each species, (iv) summed the leverages on a given checklist to provide a measure of marginal value for every checklist (i.e. the cumulative value/importance of a sampling event to inform our total knowledge of species' trends, across many species), (v) tested specific predictions (appendix 1 in the electronic supplementary material) which may influence the marginal value of a citizen science sampling event [33], and (vi) used these associations to predict the expected marginal value on a daily basis.

## 2. Methods

We tested our predictions throughout the Greater Sydney Region (approx. 12 400 km$^2$), delineating grids across the region of varying size (5, 10, 25 and 50 km$^2$), where a grid represented a 'site'. We used the R statistical environment [68] to carry out all analyses, relying heavily on the tidyverse [69], ggplot2 [70] and sf [71] packages.

In order to test our predictions, we used the eBird basic dataset (version ebd—relDec—2018; available at https://ebird.org/data/download), subsetting the data between 1 January 2010 and 31 December 2018. eBird is a successful citizen science project with greater than 600 million observations contributed by greater than 400 thousand participants, globally [15,72,73]. eBird relies on volunteer birdwatchers who collect data in the form of 'checklists'—a list of all species identified (audibly or visually) for given spatio-temporal coordinates. eBird relies on an extensive network of regional reviewers who are local experts of the avifauna [74] to ensure data quality [72].

### (a) Trend detection model

We first filtered the eBird basic dataset [51,75,76], by the following criteria: (i) we only included complete checklists, (ii) we only included terrestrial bird species, (iii) we removed any nocturnal checklists, (iv) we only included checklists which were greater than 5 min and less than 240 min in duration, (v) we only included checklists which travelled less than 5 km or covered less than 500 Ha, and (vi) we only included checklists which had greater than four species on it, as checklists with less than four species were likely to be targeted searches for particular species [58,77].

For any species with more than 50 observations ($n = 235$), we fit a generalized linear model using the 'glm' function in R, based on presence/absence with a binomial family distribution [58,60]. The models consisted of a continuous term for day, beginning 1 January 2010, and a categorical term for county, providing a spatial component to the models (e.g. figure 1). We also included an offset term for the number of species seen on a given eBird checklist, accounting for temporal and spatial effort of that

checklist [77]. This specific linear model may not be suitable for species which have varying detection probabilities throughout the full-annual cycle, but a large suite of models is possible in our framework. A total of 25 995 sampling events (i.e. eBird checklists) was used to fit each model. For the top 50 species in our analysis, we further investigated the robustness of these trend estimates in respect to sample size (appendix 2 in the electronic supplementary material).

### (b) Statistical leverage

Statistical leverage measures the influence of a particular observation on the predicted relationship between the dependent and independent variables [78]. In other words, it is a measure of how much a given observation affects the statistical model. In our instance, as is likely to be the case for all trend detection models, we had multiple predictor variables. Because for trend detection, we are interested in one specific model parameter—the temporal component—dfBeta rather than Cook's distance is appropriate [79]. dfBeta measures the change to one model parameter, after omitting the $i$th observation from the dataset [79,80]. It follows the formula

$$\text{dfBeta} = \hat{\beta} - \beta_i = \frac{(X'X)^{-1}X_i r_i}{1 - h_i},$$

where $X$ is the predictor variable matrix, $r$ the residual vector, $i$ $h$ the $i$th diagonal member and $i$ $x$ the $i$th line of matrix $X$. The value of dfBeta tends to decline with an increase in the number of observations as the trend becomes well understood.

In our case, every sampling event for each species received a dfBeta value (i.e. each species received 25 995 measures of dfBeta), using the 'dfBetas' function from R [68]. The measure of statistical leverage, then, of a given checklist was the sum of the absolute value of the dfBeta measures for each species (i.e. the sum of all 235 dfBetas). This measure of statistical leverage was thus a measure of a checklist's influence in understanding cumulative species' trends throughout the Greater Sydney Region, and accordingly represented the marginal value of that particular checklist. Failing to observe a species produces a dfBeta which can be quite important in detecting a species decline.

### (c) Parameter calculation

After our model was fit from 2010 to 2018, we calculated the predicted parameters of interest for each day in 2018 ($n = 365$). For each individual grid, at each of the grain sizes, we dynamically calculated the following parameters, related to our predictions (appendix 1 in the electronic supplementary material): (i) whether a grid cell had ever been sampled, (ii) the distance to the nearest sampled grid cell, (iii) the median sampling interval of a grid cell, (iv) the median sampling interval of the nearest sampled grid cell, (v) days since the last sample in a grid cell, (vi) the duration of sampling in a grid cell (most recent sample minus the earliest sampled date), and (vii) the number of unique sampling days within the grid cell. Note that these parameters depend on the sampling in the days prior to that particular observation and do not consider the sampling in subsequent days.

We then subsetted the leverage calculations (see above) for each of the days in 2018, given we knew where people sampled, relative to the parameters for each of the grids on that day. We ran a linear regression for each of the different grain sizes considered in the analysis to investigate which parameters could forecast checklist influence. Prior to modelling, duration was highly correlated with median sampling interval for the majority of the grain size analyses, and as such, was excluded from consideration. Given the parameters' correlation varied among grain sizes (appendix 3 in the electronic supplementary

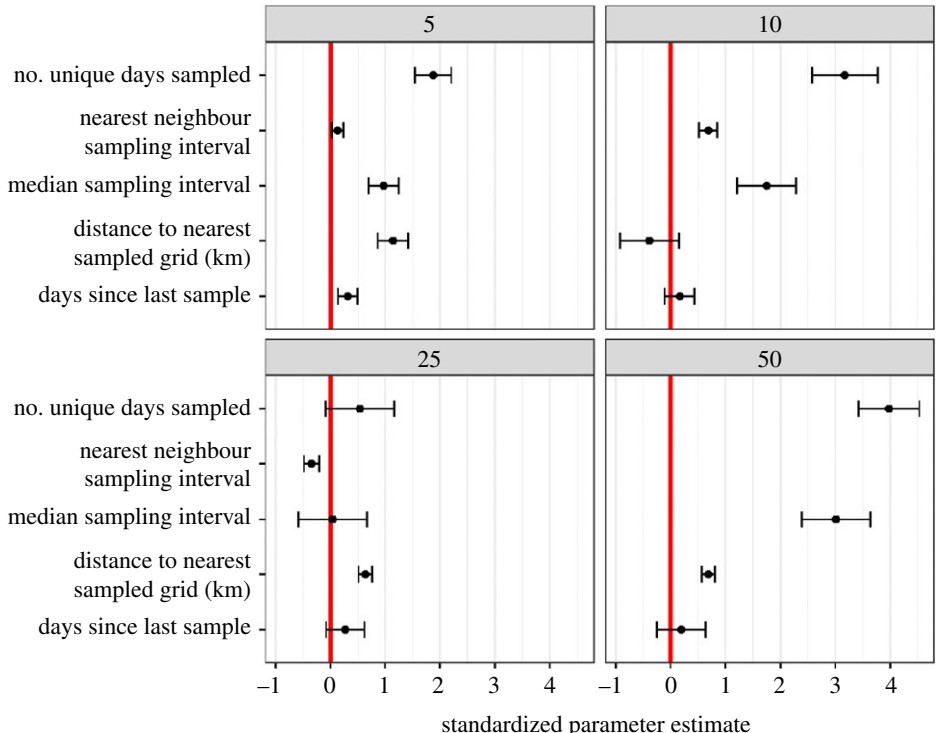

**Figure 2.** The parameter estimates (and 95% confidence intervals) for four separate linear models (i.e. at each of the respective grain sizes), showing the relative strength of the results and effect sizes for each of the predictors. Full summary statistics for each of the predictors can be found in appendix 4 in the electronic supplementary material. Variables were log-transformed and then standardized, allowing for direct comparison among effect sizes. (Online version in colour.)

material), we needed to ensure a robust, and simple model. All variables were log-transformed and then standardized prior to modelling, ensuring that the effect sizes of the given parameters were meaningful. The response variable, dfBeta (i.e. marginal value) was log-transformed prior to modelling to meet model assumptions. Thus, the final model included a log-transformed dfBeta response variable, regressed against log-transformed standardized median sampling interval, number of days sampled, days since the last sample, distance to the nearest sampled neighbour, and the neighbour's median sampling interval.

After our model was fitted, we used the 'augment' function from the broom package [81] to predict the expected leverage for every grid cell in the Greater Sydney Region, for every day. For grid cells which were unsampled, we assigned them the mean of the sampled grid cells, based on our lack of evidence that unsampled cells were significantly more valuable than sampled cells. Where one grid had multiple predicted leverages (i.e. where a grid had more than one checklist in a day), we randomly sampled to one of these expected leverages. This prediction process was repeated for every day of 2018.

## 3. Results and discussion

### (a) Tests of predictions

We found weak evidence that visiting an unsampled site was marginally more valuable than visiting an already sampled site, but we did find that as grain size increases the importance of sampling unsampled sites also increases. There was no statistical clarity for the 5 km ($p = 0.669$; effect size $= -0.25 \pm 0.58$) and 10 km ($p = 0.093$; effect size $= 1.27 \pm 0.76$) grain sizes, but there was for the 25 km ($p = 0.035$; effect size $= 2.98 \pm 1.42$) grain size. At the 50 km grain size, this test was not possible because all sites had been sampled. These results suggest that stratified sampling—an approach which aims for equal sampling among sites [38,57]—is not

necessarily the most appropriate approach for detecting trends using citizen science data. In other words, citizen scientists are likely already sufficiently sampling biodiversity in space: they appropriately identify and sample 'hotspots' in space that should receive the most sampling attention. But the effect of citizen scientists visiting 'popular' locations (e.g. spots known for their bird diversity) could exclude the discovery of other known 'hotspots' in the same region.

For those sampled sites, however, we found a generally positive relationship between a number of our predicted parameters (detailed predictions can be found in appendix 1 in the electronic supplementary material) and the marginal value of a sampling event (figure 2). Full summary statistics for each of our predictors can be found in appendix 4 in the electronic supplementary material, but the range, median and interquartile range, respectively, can be found in parentheses after the referenced parameter. The number of unique days sampled (5 km: 1–1222, 25, 228; 10 km: 1–1894, 79, 424; 25 km: 1–2946, 1103, 2071.5; 50 km: 11–3004, 2284, 808)—probably represented from known 'hotspots' identified by citizen scientists—had the strongest, positive, effect size, and this was robust to grain size comparisons. The median sampling interval (5 km: 1–2450, 31, 124; 10 km: 1–1401, 11, 57; 25 km: 1–821, 1, 5; 50 km: 1–193, 1, 0) was also strongly associated with high value samples, with an exception at the 25 km grain size. Distance to the nearest sampled site (5 km: 1.9–19.7, 5, 2.1; 10 km: 0–23.1, 10, 0.7; 25 km: 14.8–25, 22, 4.3; 50 km: 32.3–45.1, 37.5, 3.6) and the nearest-neighbour sampling interval (5 km: 1–2450, 43.5, 170; 10 km: 1–1401, 14, 54; 25 km: 1–821, 2, 5; 50 km: 1–1, 1, 0) influenced the value of a sampling event less than the other parameters. Surprisingly, the number of days since the last sample (5 km: 1–2935, 39, 228; 10 km: 1–1645, 10, 86; 25 km: 1–693, 1, 4; 50 km: 1–316, 1, 0), while positively associated, had less influence than other parameters. See figure 2 for

standardized parameter estimates (i.e. effect sizes). The fact that days since last sample had a lower effect size than both the median sampling interval and the number of unique days sampled, highlights the value of 'revisiting' a site (i.e. 'hotspot') in order to extract the maximum amount of information.

The 'history' of a site is particularly important while considering whether to sample that site: the number of unique days sampled was the strongest predictor for all but the 25 km (second strongest) grain size, suggesting that observations from sites with a long time series are relatively more valuable. Because sites with larger median sampling intervals were positively associated with the marginal value of a citizen science observation, a secondary goal could be to decrease (i.e. left-shift the distribution) the median sampling interval of sites by targeting sites with the largest median sampling intervals; providing some structure to unstructured citizen science projects.

We found generally consistent results, albeit with variation in parameters: no predictor was consistently significant across all grain sizes. Nevertheless, our findings appear to be robust to spatial scale, at least within a regional level. It is critical to track biodiversity trends at multiple spatial scales [6], as biodiversity estimates sometimes change dependent on the spatial scale [40]. In comparison with other regions in Australia, the distribution of sampled grids in Sydney is generally similar—many grids unsampled or sampled only a few times, and then large variation among the rest of the grids (appendix 5 in the electronic supplementary material). Different regions have the same underlying 'starting point' in the current sampling regime, suggesting our results are generalizable among regions. Although this may only be applicable at a regional scale, and future work should further investigate these patterns at large, continental scales, where the grain size is proportional to the spatial scale of the study. For example, within all of Australia, it is likely that unsampled regions will be significantly more important because there are many 'gaps' in the data, and effort could thus be directed from well sampled regions to unsampled regions.

## (b) Applications of our predictions

Providing dynamic feedback to citizen science participants has proved successful for many citizen science projects [44,82,83]. This feedback is generally in the form of leaderboards, presenting the number of submissions or number of unique species someone has contributed [84]. But leaderboards tend to focus on outputs, incentivizing finding rather than looking, leading to perverse outcomes related to the redundancies and gaps in citizen science data. We sought to develop an outcome-based incentive by using our fitted models to predict the expected value of a given citizen science observation, dynamically, for any given day (e.g. figure 3; https://github.com/coreytcallaghan/optimizing-citizen-science-sampling/blob/master/Figures/dynamic_map.gif). This approach required us to analyse data from the past first, using a model with all observations for 2018, based on statistical leverage calculated from 2010 to 2018, in order to predict the expected marginal value for any given day. We envision a dynamic approach (https://github.com/coreytcallaghan/optimizing-citizen-science-sampling/blob/master/Figures/dynamic_map.gif) in future citizen science projects, which would ultimately guide participants to sites which should be sampled on any given day—or in

a given week, month or year. In this instance, leaderboards would move past numbers of species or submissions and could be derived based on a participant's cumulative value to the citizen science dataset. Instead of participants preferentially chasing specific species, this approach would guide participants to the sites with the highest expected marginal value for the biodiversity dataset. For example, we imagine visitor centres across the world at national parks or urban greenspaces providing their visitors on any given day a localized map showing which trail someone should visit if they are interested in contributing the greatest value to that park's biodiversity knowledge, through citizen science. The global pull of ecotourism [85] is increasing exponentially, creating the potential for people to contribute to local biodiversity knowledge in areas that are traditionally undersampled, and with this framework, the collective effort of citizen scientists can be maximized.

Another critical component to efficiently direct effort and maximize the collective effort of citizen scientists is by understanding critical thresholds necessary for reliable estimates of trend detection. If the minimum number of sampling events for a region is understood, then citizen science effort could appropriately be directed to areas where these critical thresholds are not yet met. We preliminarily found that for the top 50 species in our analysis, approximately 11 700 checklists were necessary for a 50% reduction/convergence in the slope estimate based on our model (appendix 2 in the electronic supplementary material). This result is comparable with a study in the USA which found that approximately 10 000 eBird checklists were necessary to provide reliable trend estimates [60]. Future work should investigate critical thresholds for biodiversity analyses and how these interact with efficiently directing citizen science effort.

We focused our framework on a specific statistical outcome: trend detection. Many other ecological outcomes arise from citizen science datasets, including species distribution models [47,48], phylogeographical research [55,56], invasive species detection [53,54] or phenological research [51,52]. Each potential outcome will have different optimal sampling strategies in space and time, probably with nuanced trade-offs between outcomes. For example, an intended outcome of a species distribution model would be likely to place greater value on observations from unsampled sites [86] than for species trend detection. But these different outcomes can still be quantified in the same framework we introduce here: this framework could be applied to a wide suite of statistical models—including for different taxa and including more complicated trend analysis accounting for intra-annual varying detection probabilities. The key piece of information is some form of statistical leverage that can be calculated from a potential statistical model.

## (c) Conclusions

Since eBird's inception in 2002, citizen scientists have collectively contributed greater than 30 million effort hours. And this is only one citizen science project, focused on birds. Our approach should be tested for other taxonomic groups, ensuring generalizability. Clearly, the cumulative effort put forth by citizen scientists is immense; arguably, citizen science will continue to shape the future of ecology and conservation—as it has substantially for the past couple of centuries [63]—with an increasingly critical role in

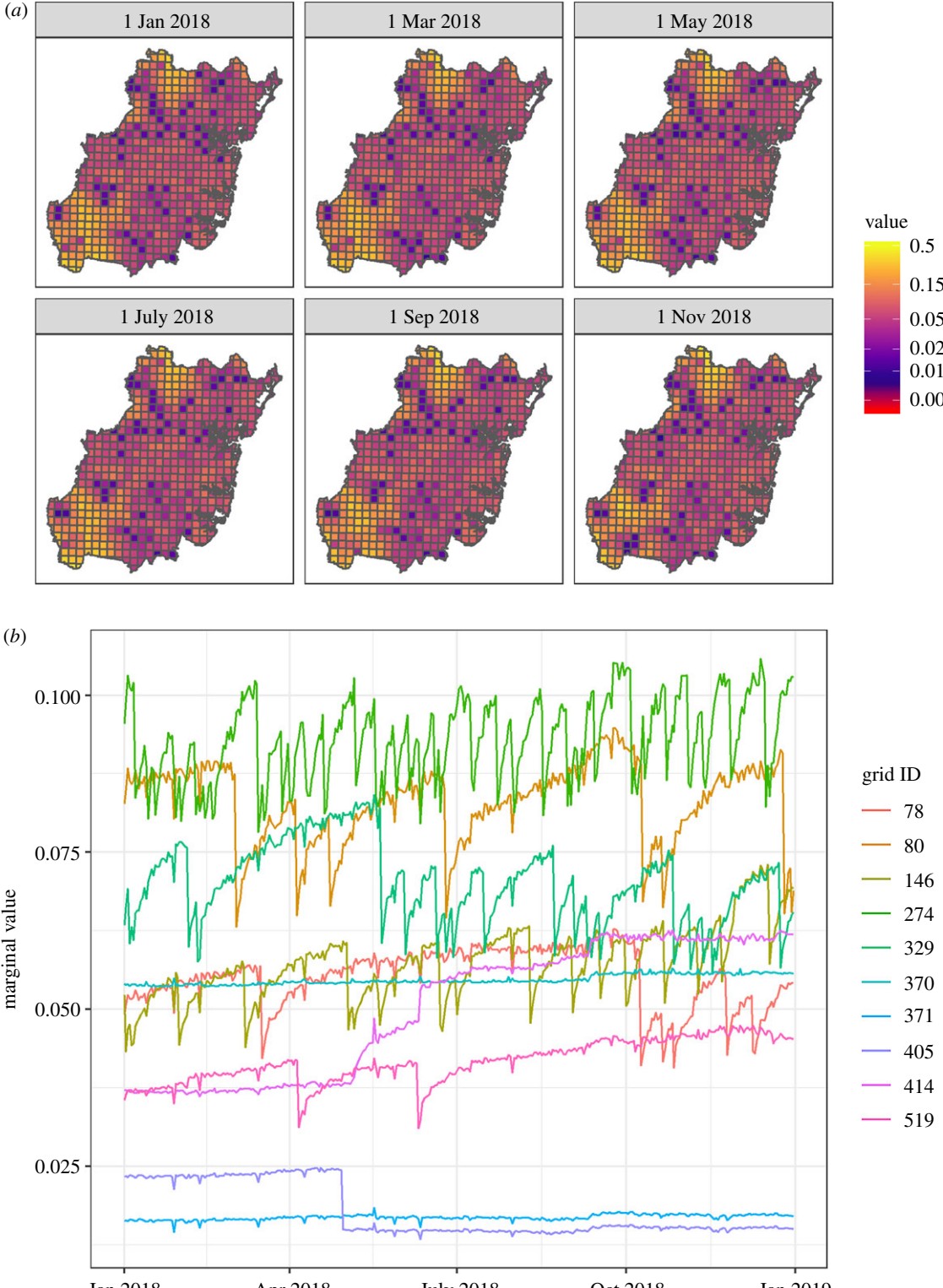

**Figure 3.** (*a*) A map of predicted expected marginal value for six different days in 2018, throughout the Greater Sydney Region, showing the highest valued sites that would optimize the collective knowledge on biodiversity trends throughout the Greater Sydney Region. This prediction step is dynamic: predictions are updated as new observations are submitted to the citizen science database. Expected marginal value maps will need to be updated fast in parts of the world where sampling rates are high, but this can be done at a slower rate where sampling is less frequent, and this will also vary among projects. (*b*) The changes in expected marginal value for 2018 for 10 randomly chosen grid cells at the 5 km$^2$ grid size. Some sites' expected marginal values remain relatively constant through time (e.g. grid 371) while others are highly variable (e.g. grid 274), and others undergo distinct step-changes (e.g. grid 405) corresponding to whether that grid was recently sampled or not. This is a dynamic approach, showing how the expected marginal value of a grid changes based on our parametrized model results (https://github.com/coreytcallaghan/optimizing-citizen-science-sampling/blob/master/Figures/dynamic_map.gif). (Online version in colour.)

monitoring of biodiversity [20,65]. But we need to look towards the future. Are there mechanisms we can put in place now which will increase our collective knowledge gleaned from citizen science datasets for biodiversity in the

future? We highlight general rules which could help guide citizen science participants to better sampling in space and time: the number of unique days sampled and the largest median sampling intervals both positively correlate with

the marginal value of a citizen science observation. Moreover, we demonstrate a framework which citizen science projects can implement to better optimize their sampling designs, which can be tailored to specific citizen science project goals.

Data accessibility. All eBird data are freely available for download (https://ebird.org/data/download), but the necessary portion of the eBird basic dataset (i.e. for the Greater Sydney Region), along with spatial data and code to reproduce our analyses, are available at: https://doi.org/10.5281/zenodo.3402307.

Authors' contributions. C.T.C., W.K.C., A.G.B.P., J.J.L.R. and R.E.M. conceived the study. C.T.C. and W.K.C. carried out the analysis and wrote the first draft of the manuscript. All authors contributed to editing and revising the manuscript.

Competing interests. We declare we have no competing interests.

Funding. We received no funding for this study.

Acknowledgements. We thank the countless eBird contributors who are continuously making open-access bird observation data available, and the eBird team at the Cornell Lab for curating this valuable dataset. We also thank two anonymous reviewers who helped improve the manuscript.

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
