## [Reviewer comments · Proceedings of the Royal Society B: Biological Sciences]

Review History

RSPB-2019-1487.R0 (Original submission)

Review form: Reviewer 1 (David M. Watson)

Recommendation

Accept with minor revision (please list in comments)

Scientific importance: Is the manuscript an original and important contribution to its field?

Excellent

General interest: Is the paper of sufficient general interest?

Excellent

Quality of the paper: Is the overall quality of the paper suitable?

Excellent

Is the length of the paper justified?

Yes

Should the paper be seen by a specialist statistical reviewer?

No

Do you have any concerns about statistical analyses in this paper? If so, please specify them explicitly in your report.

No

It is a condition of publication that authors make their supporting data, code and materials available - either as supplementary material or hosted in an external repository. Please rate, if applicable, the supporting data on the following criteria.

Is it accessible?

Yes

Is it clear?

Yes

Is it adequate?

Yes

Do you have any ethical concerns with this paper?

No

Comments to the Author

In this contribution, Callaghan and colleagues explore the utility of unstructured citizen-scientist-collected occurrence data for detecting population trends, evaluate which factors affect the inferential impact of particular records, and suggest ways to optimize data collection to maximise inferential power. There is a lot to like about this work – it is thorough, uses carefully-designed analyses to address the stated objectives, and considers the broader implications of their findings.

I have one substantive comment on this contribution:

The authors use bird checklists collected in the greater Sydney area as a case study. In addition to being a relatively well studied group of organisms, this is also an area with both a high density of active citizen scientists and a large number of species. In considering the wider utility of the approach they are suggesting, it begs the question – how representative is this region / this dataset for global citizen science initiatives? Further, is there a minimum number of checklists / active observers for reliable trends to be inferred. Rather than a criticism of their work, this is about being more circumspect about those taxa / regions best suited to their approach.

They could address this in several ways. One would be to use a rarefaction-based approach to remove checklists and explore the effect of sampling density / frequency / number of observers on trend analysis to evaluate the shape of the relationship and identify any critical thresholds. They could also compare their Sydney dataset with somewhere else. Australia is the ideal place to explore this, with very high population density in a handful of coastal capital cities, and very low population densities elsewhere. Thus, rather than optimizing where data are collected within a particular region to detect trends, it may be that simply acquiring more data ANYWHERE in the region is the priority, attracting more effort from adjacent, better sampled areas.

By expanding the scope of their work, not only will they be better placed to evaluate the broader utility of their approach, it will also integrate their work with the extensive literature on sampling effort determination

Review form: Reviewer 2 (Michael Pocock)

Recommendation

Accept with minor revision (please list in comments)

Scientific importance: Is the manuscript an original and important contribution to its field?

Excellent

General interest: Is the paper of sufficient general interest?

Good

Quality of the paper: Is the overall quality of the paper suitable?

Excellent

Is the length of the paper justified?

Yes

Should the paper be seen by a specialist statistical reviewer?

No

Do you have any concerns about statistical analyses in this paper? If so, please specify them explicitly in your report.

No

It is a condition of publication that authors make their supporting data, code and materials available - either as supplementary material or hosted in an external repository. Please rate, if applicable, the supporting data on the following criteria.

Is it accessible?

Yes

Is it clear?

Yes

Is it adequate?

Yes

Do you have any ethical concerns with this paper?

No

Comments to the Author

I believe that this is an important paper that addresses some crucial issues for citizen science, and does so in a clear way. I too am exploring these issues of making citizen science more 'adaptive' by incentivising recorders to make visits at the times and places where their records will be more informative. This paper is a good contribution to this and (I think) the first published example. I like the fact that this paper simplifies this potential complex analysis to some 'rules of thumb' about what is most important and makes some specific recommendations.

The Introduction is very well-written.

Methods:

I think it would be helpful in the Methods to have an overview of what you did. I had to re-read the Methods several times to fully understand exactly what you did. I think you: 1. took 25995

checklists and calculated linear trends for 235 species; 2. calculated the statistical leverage for the 25995 x 235 combinations of species observations; 3. summed the leverages for each checklist to give an overall importance for each checklist; 4. related the importance of the checklists to their traits; 5. predicted the expected leverage on a daily basis for grid cells in the region.

L125 please note that you modelled a linear trend with your logistic regression. This does not account for seasonality in species (or their detection). It is not suitable for long-term trends which are unlikely to be linear. This should be addressed in the Discussion because your framework could be applied to other model outputs.

L133 Please rephrase "the probability of observation of that species with respect to time" for clarity.

L140-143 suggest change "given" to "each" for clarity.

L164 These correlations should be given in an Appendix.

I suggest that you move the hypotheses into the main text (if you have space), ideally with the summary statistics for the parameters.

Results and Discussion

You brought out some simple messages from this analysis nicely in the Discussion, e.g. the value of sampling new locations v revisiting locations.

L193 Really interesting thought. Is there circularity here: revisited sites are valuable because they provide much information, hence they appear to be hotspots, hence recorders are "already sampling the 'hotspots'"?

L187-190 Can you please include the effect sizes. Actually, I wonder if visiting new sites becomes increasingly important as the grain size increases? (i.e. visiting a new 5km is not valuable, visiting a new 25km square is valuable.)

L197ff You have done a good job of summarising the results, but I think you should acknowledge that there is quite a lot of variation in these predictors (cp L217 "generally consistent"). No predictor is consistently significant across all grain sizes.

L210 Please provide some summary statistics about each of the predictors (e.g. range, inter-quartile range and median)

L212 Interestingly, this recommendation (to 'left shift' the distribution) is a form of structuring.

L213ff If the inflation effect of the history of the site is true, then I'd expect this to be consistent. But the effect of number of days sampled is not consistent at the next largest grain size (25km). Can you explain this discrepancy?

L217 Pedantically, this is not the size of the 'site', but the grid size used for aggregation and analysis.

L234 "look backward first". Please clarify this - it was not immediately obvious what you meant. I think you mean "firstly analysed past data... in order to make predictions during the following year",

L242 and Fig 3. You use 'expected values' - but of what? - I suggest selecting a more descriptive term.

Fig 2. Please expand the legend so that it is fully understandable on its own.

Fig S1. This is a valuable figure that could be brought into the main text, if there is space. What is 'marginal value'? You mention it in L146, but I think it is a synonym for 'leverage' and 'expected value'? I prefer using one term for one idea.

I thought that the Discussion was strong in terms of the practical recommendations and next steps.

Decision letter (RSPB-2019-1487.R0)

16-Aug-2019

Dear Mr Callaghan:

Your manuscript has now been peer reviewed and the reviews have been assessed by an Associate Editor. The reviewers' comments (not including confidential comments to the Editor) and the comments from the Associate Editor are included at the end of this email for your reference. As you will see, the reviewers and the Editors have raised some concerns with your manuscript and we would like to invite you to revise your manuscript to address them.

Research ethics:

Use of animals and field studies:

It is a condition of publication that you make available the data and research materials supporting the results in the article. Datasets should be deposited in an appropriate publicly available repository and details of the associated accession number, link or DOI to the datasets must be included in the Data Accessibility section of the article

(<https://royalsociety.org/journals/ethics-policies/data-sharing-mining/>). Reference(s) to datasets should also be included in the reference list of the article with DOIs (where available).

Please submit a copy of your revised paper within three weeks. If we do not hear from you within this time your manuscript will be rejected. If you are unable to meet this deadline please let us know as soon as possible, as we may be able to grant a short extension.

Best wishes,

Dr Sasha Dall
mailto: proceedingsb@royalsociety.org

Associate Editor

Board Member: 1

Comments to Author:

Both reviewers are very positive about this manuscript addressing ways to optimise biodiversity sampling in citizen science schemes. They make valuable recommendations which it would be useful for the authors to address, in order to clarify their approach and to explore fully the scope of their results.

Reviewer(s)' Comments to Author:

Referee: 1

Comments to the Author(s)

In this contribution, Callaghan and colleagues explore the utility of unstructured citizen-scientist-collected occurrence data for detecting population trends, evaluate which factors affect the inferential impact of particular records, and suggest ways to optimize data collection to maximise inferential power. There is a lot to like about this work – it is thorough, uses carefully-designed analyses to address the stated objectives, and considers the broader implications of their findings.

I have one substantive comment on this contribution:

The authors use bird checklists collected in the greater Sydney area as a case study. In addition to being a relatively well studied group of organisms, this is also an area with both a high density of active citizen scientists and a large number of species. In considering the wider utility of the approach they are suggesting, it begs the question – how representative is this region / this dataset for global citizen science initiatives? Further, is there a minimum number of checklists / active observers for reliable trends to be inferred. Rather than a criticism of their work, this is about being more circumspect about those taxa / regions best suited to their approach.

They could address this in several ways. One would be to use a rarefaction-based approach to remove checklists and explore the effect of sampling density / frequency / number of observers on trend analysis to evaluate the shape of the relationship and identify any critical thresholds. They could also compare their Sydney dataset with somewhere else. Australia is the ideal place to explore this, with very high population density in a handful of coastal capital cities, and very low population densities elsewhere. Thus, rather than optimizing where data are collected within a particular region to detect trends, it may be that simply acquiring more data ANYWHERE in the region is the priority, attracting more effort from adjacent, better sampled areas.

By expanding the scope of their work, not only will they be better placed to evaluate the broader utility of their approach, it will also integrate their work with the extensive literature on sampling effort determination

Referee: 2

Comments to the Author(s)

I believe that this is an important paper that addresses some crucial issues for citizen science, and does so in a clear way. I too am exploring these issues of making citizen science more 'adaptive' by incentivising recorders to make visits at the times and places where their records will be more informative. This paper is a good contribution to this and (I think) the first published example. I like the fact that this paper simplifies this potential complex analysis to some 'rules of thumb' about what is most important and makes some specific recommendations.

The Introduction is very well-written.

Methods:

I think it would be helpful in the Methods to have an overview of what you did. I had to re-read the Methods several times to fully understand exactly what you did. I think you: 1. took 25995 checklists and calculated linear trends for 235 species; 2. calculated the statistical leverage for the 25995 x 235 combinations of species observations; 3. summed the leverages for each checklist to

give an overall importance for each checklist; 4. related the importance of the checklists to their traits; 5. predicted the expected leverage on a daily basis for grid cells in the region.

L125 please note that you modelled a linear trend with your logistic regression. This does not account for seasonality in species (or their detection). It is not suitable for long-term trends which are unlikely to be linear. This should be addressed in the Discussion because your framework could be applied to other model outputs.

L133 Please rephrase "the probability of observation of that species with respect to time" for clarity.

L140-143 suggest change "given" to "each" for clarity.

L164 These correlations should be given in an Appendix.

I suggest that you move the hypotheses into the main text (if you have space), ideally with the summary statistics for the parameters.

Results and Discussion

You brought out some simple messages from this analysis nicely in the Discussion, e.g. the value of sampling new locations v revisiting locations.

L193 Really interesting thought. Is there circularity here: revisited sites are valuable because they provide much information, hence they appear to be hotspots, hence recorders are "already sampling the 'hotspots'"?

L187-190 Can you please include the effect sizes. Actually, I wonder if visiting new sites becomes increasingly important as the grain size increases? (i.e. visiting a new 5km is not valuable, visiting a new 25km square is valuable.)

L197ff You have done a good job of summarising the results, but I think you should acknowledge that there is quite a lot of variation in these predictors (cp L217 "generally consistent"). No predictor is consistently significant across all grain sizes.

L210 Please provide some summary statistics about each of the predictors (e.g. range, inter-quartile range and median)

L212 Interestingly, this recommendation (to 'left shift' the distribution) is a form of structuring.

L213ff If the inflation effect of the history of the site is true, then I'd expect this to be consistent. But the effect of number of days sampled is not consistent at the next largest grain size (25km). Can you explain this discrepancy?

L217 Pedantically, this is not the size of the 'site', but the grid size used for aggregation and analysis.

L234 "look backward first". Please clarify this - it was not immediately obvious what you meant. I think you mean "firstly analysed past data... in order to make predictions during the following year",

L242 and Fig 3. You use 'expected values' - but of what? - I suggest selecting a more descriptive term.

Fig 2. Please expand the legend so that it is fully understandable on its own.

Fig S1. This is a valuable figure that could be brought into the main text, if there is space. What is 'marginal value'? You mention it in L146, but I think it is a synonym for 'leverage' and 'expected value'? I prefer using one term for one idea.

I thought that the Discussion was strong in terms of the practical recommendations and next steps.

Author's Response to Decision Letter for (RSPB-2019-1487.R0)

See Appendix A.

Decision letter (RSPB-2019-1487.R1)

06-Sep-2019

Dear Mr Callaghan

I am pleased to inform you that your manuscript RSPB-2019-1487.R1 entitled "Optimizing future biodiversity sampling by citizen scientists" has been accepted for publication in Proceedings B.

The BM has recommended publication, but also suggests some minor revisions to your manuscript. Therefore, I invite you to respond to the comments and revise your manuscript. Because the schedule for publication is very tight, it is a condition of publication that you submit the revised version of your manuscript within 7 days. If you do not think you will be able to meet this date please let us know.

Sincerely,

Dr Sasha Dall
Editor, Proceedings B
<mailto:proceedingsb@royalsociety.org>

Associate Editor:

Board Member

Comments to Author:

I was impressed with the thorough measures taken by the authors to implement the recommendations of the two referees, and agree that the additional supplementary material provides valuable context for the results. I have two minor recommendations - one concerning format, and the other concerning clarification of the effects of one variable.

1. References should be numbered in citation order rather than alphabetical order.

2. Lines 220-237: please check that the units and directions of effects are clear in these summary statistics. Within this section (and elsewhere), the main effect I would like to see clarified is that of median sampling interval (also referred to as median waiting time in Appendix 3). In Appendix 1 you hypothesise that, "the median sampling interval would be positively associated with the

value of a citizen science observation; i.e., observations from sites with high median waiting times would be more valuable" but from your results I understand that the median sampling interval had a negative (shorter intervals, stronger value) effect, because you later state, "A secondary goal should be to decrease (i.e., left-shift the distribution) the median sampling interval of sites; providing some structure to unstructured citizen science projects." (lines 242-243). I might however have misunderstood this point, as your conclusion states "largest median sampling intervals both positively correlate with the relative marginal value" (lines 321-322) – so please make sure the descriptions of this effect are clear and unambiguous.

Author's Response to Decision Letter for (RSPB-2019-1487.R1)

See Appendix B.

Decision letter (RSPB-2019-1487.R2)

09-Sep-2019

Dear Mr Callaghan

I am pleased to inform you that your manuscript entitled "Optimizing future biodiversity sampling by citizen scientists" has been accepted for publication in Proceedings B.

Your article has been estimated as being 9 pages long. Our Production Office will be able to confirm the exact length at proof stage.

Open Access

Paper charges

Sincerely,

Editor, Proceedings B
mailto:proceedingsb@royalsociety.org

Appendix A

Professor Spencer Barret, FRS
Department of Ecology and Evolutionary Biology
University of Toronto
Editor, The Royal Society Proceedings B

September 2nd, 2019

Dear Professor Barret,

On behalf of my co-authors and myself, I would like to formally resubmit our paper, entitled “*Optimizing future biodiversity sampling by citizen scientists*” for publication in *The Royal Society Proceedings B*.

We thank you for your consideration of the manuscript and the constructive reviews which helped us significantly improve this paper. We have listed all the referees’ comments below and describe how we have responded to each suggestion.

We hope that the revisions to the manuscript meet with your satisfaction, making this a valuable contribution to *The Royal Society Proceedings B*.

Sincerely,
Corey Callaghan

Centre for Ecosystem Science
School of Biological, Earth and Environmental Sciences
UNSW Sydney
E: c.callaghan@unsw.edu.au
P: +61 421 601 388

Associate Editor
Board Member: 1
Comments to Author:

Both reviewers are very positive about this manuscript addressing ways to optimise biodiversity sampling in citizen science schemes. They make valuable recommendations which it would be useful for the authors to address, in order to clarify their approach and to explore fully the scope of their results.

Response:

We address each of the referees' comments below. We have also made minor text changes to meet with the page limit (e.g., have reduced the abstract to < 200 words). All changes in response to the reviewers are copied below, but we also uploaded an additional file with all tracked changes.

Referee: 1

Comments to the Author(s)

In this contribution, Callaghan and colleagues explore the utility of unstructured citizen-scientist-collected occurrence data for detecting population trends, evaluate which factors affect the inferential impact of particular records, and suggest ways to optimize data collection to maximise inferential power. There is a lot to like about this work—it is thorough, uses carefully-designed analyses to address the stated objectives, and considers the broader implications of their findings.

Response: *We appreciate this positive response and thank you for these suggestions which has strengthened our work.*

I have one substantive comment on this contribution:

The authors use bird checklists collected in the greater Sydney area as a case study. In addition to being a relatively well studied group of organisms, this is also an area with both a high density of active citizen scientists and a large number of species. In considering the wider utility of the approach they are suggesting, it begs the question—how representative is this region / this data-set for global citizen science initiatives? Further, is there a minimum number of checklists / active observers for reliable trends to be inferred. Rather than a criticism of their work, this is about being more circumspect about those taxa / regions best suited to their approach.

They could address this in several ways. One would be to use a rarefaction-based approach to remove checklists and explore the effect of sampling density / frequency / number of observers on trend analysis to evaluate the shape of the relationship and identify any critical thresholds. They could also compare their Sydney dataset with somewhere else. Australia is the ideal place to explore this, with very high population density in a handful of coastal capital cities, and very low population densities elsewhere. Thus, rather than optimizing where data are collected within a particular region to detect trends, it may be that simply acquiring more data ANYWHERE in the region is the priority, attracting more effort from adjacent, better sampled areas.

By expanding the scope of their work, not only will they be better placed to evaluate the broader utility of their approach, it will also integrate their work with the extensive literature on sampling effort determination

Response: *These are all excellent comments and suggestions. We are currently working on some of these suggestions as parts of different manuscripts (e.g., a manuscript which is investigating trend analysis in randomly sampled polygons throughout Australia including and excluding potential false positives in the data). We have explored a variety of these suggestions in our revision, and believe this has strengthened our manuscript. To address these comments/suggestions we quote the reviewer below and respond to different parts of his/her comments in more detail – not necessarily in order.*

- **REVIEWER 1:** “In addition to being a relatively well studied group of organisms...”

Response: *We chose birds as an example taxon as they are relatively well-studied, and benefit from a long history of citizen science sampling (e.g., Christmas Bird Counts, Breeding Bird Surveys in the US and UK). The bias towards birds in citizen science data (Mair and Ruete 2016) is well-known. We also acknowledge that eBird is the most successful citizen science project in the world. Although we only looked at birds/eBird here, we see no reason to believe that these approaches would not be applicable to other taxa, and as other citizen science projects (e.g., eButterfly, FrogID, U.K. Butterfly Monitoring Scheme) continue to provide increasing amounts of data we hope that these approaches can be tested with other taxa. We are currently exploring some of these questions in a different manuscript, as we agree that they are valuable questions (e.g., using FrogID data) and addressing these in detail is outside the focus of the current manuscript, and we are currently at our maximum page limit. In order to address this comment in the manuscript, we are more circumspect and directly acknowledge that our approach is only currently applicable to birds in the ‘conclusions’ paragraph, by saying (lines 312 – 313):*

“And this is only one citizen science project, focused on birds. Our approach should be tested for other taxonomic groups, ensuring generalizability.”.

Further, different statistical models may need to be employed for different taxa, which is a fundamental part of our approach’s generalizability. We therefore highlight in the revised manuscript that different taxa may require different statistical models (lines 303 – 308):

“But these different outcomes can still be quantified in the same framework we introduce here: this framework could be applied to a wide suite of statistical models – including for different taxa and including more complicated trend analysis accounting for intra-annual varying detection probabilities. The key piece of information is some form of statistical leverage that can be calculated from a potential statistical model.”.

- **REVIEWER 1:** “... this is also an area with both a high density of active citizen scientists and a large number of species”

Response: *We see no reason that the number of species in an area would influence the generalizability of our results, because our approach is not species-focused per se, but rather focused on sampling in space and time, taking a cumulative measure of a sampling event on X number of species in an area. But we do agree that this is an active area of citizen scientists, with a relatively large number of species. However, we believe that our approach may be best suited at a regional scale – i.e., it is unlikely and potentially unethical (e.g., carbon emissions) to encourage citizen scientists to regularly sample outside of their home region, unless they are on a specific trip (which happens often in birding). We do further acknowledge the aspect of density of citizen scientists and the regional versus continental scale in our revised manuscript. For a more thorough response to this comment and adjusted text in the revised manuscript, see the next bullet point.*

- **REVIEWER 1:** “In considering the wider utility of the approach they are suggesting, it begs the question—how representative is this region / this data-set for global citizen science initiatives?” & “They could also compare their Sydney dataset with somewhere

else. Australia is the ideal place to explore this, with very high population density in a handful of coastal capital cities, and very low population densities elsewhere.”

Response: We believe that the approach should be applicable to other regions in Australia, and the world, if the ‘starting point’ from which sampling is taking place is similar. In other words, if the distribution of the number of samples per grid in a region is similar, then our approach should work because the sampling regimes are at least somewhat similar initially. To test this, we have now incorporated 15 cities in Australia and collected all eBird checklists within a 50 km buffer (we chose a mix of large coastal cities and small inland cities). (This approach is slightly different to our analysis presented in the main text because we had a shapefile of the Greater Sydney Region, but we did not have access to such shapefiles for other cities.) We then gridded each 50 km buffer into 5 km grids and counted the number of eBird checklists per grid:

Clearly, the distribution of grid sampling throughout each region is relatively similar: many grids are unsampled or sampled a few times, and then few grids are sampled more times for the majority of cities. Sydney is especially similar to Melbourne and Brisbane – two other large cities with lots of species and similar density of citizen scientists. We therefore feel confident that our approach is generalizable to other regions throughout Australia. We have added this figure as a supplementary figure (Appendix 5) and have further elaborated on it in the text of the manuscript (lines 247 – 256):

“Nevertheless, our findings appear to be robust to spatial scale, at least within a regional level. It is critical to track biodiversity trends at multiple spatial scales [64], as biodiversity estimates sometimes change dependent on the spatial scale [14]. In comparison with other regions in Australia, the distribution of sampled grids in Sydney is generally similar – many grids unsampled or sampled only a few times, and then large variation among the rest of the grids (Appendix 5). Different regions have the same underlying ‘starting point’ in the current sampling regime, suggesting our results are generalizable among regions. Although this may only be applicable at a regional scale, and future work should further investigate these patterns at large continental-scales where the grain size is proportional to the spatial scale of the study.”

- **REVIEWER 1:** “Further, is there a minimum number of checklists / active observers for reliable trends to be inferred.” & “One would be to use a rarefaction-based approach to remove checklists and explore the effect of sampling density / frequency / number of observers on trend analysis to evaluate the shape of the relationship and identify any critical thresholds.”

Response: *This is a good question. Indeed, the investigation of spatial rarefaction is a growing area of research interest (Bacaro et al. 2012), and offers another avenue of exploration in the future, to interact with our framework. If critical thresholds (i.e., minimum number of biodiversity sampling events) for biodiversity sampling can be identified, then effort could be directed to locations which have yet to meet these critical thresholds. We believe this is an additional and potentially very interesting component of our current analysis. Indeed, we are working on an analysis which estimates the number of eBird checklists necessary for reliable species richness estimation throughout Australia (similar to Callaghan et al. 2017).*

Nevertheless, we have now added some rarefaction analyses to the current manuscript, which we believe strengthens it substantially. Previous work in the USA has found that in comparison with BBS trends, ~ 10,000 eBird checklists are necessary to provide ‘reliable’ trend estimates (Horns et al. 2018). We did not have any ‘structured’ dataset in the Sydney region to compare our results with. Thus, we performed a ‘rarefaction’ type approach (e.g., sensitivity/power analysis), as suggested by the Reviewer. We carried out this analysis for the top 50 species in our dataset only as we believe these preliminary results are sufficient to address the question, and this analysis is tangential to the main thrust of our current manuscript. We randomly subsampled the potential pool of eBird checklists (N=25,995) from 10% to 100% of the checklists, in 5% increments. We performed 100 runs of the same GLM as presented in the main manuscript, at each of these 19 levels, for each species; fitting a total of 95,000 models. We removed any models that did not converge and whose slope

estimates were $2 SD > \text{the mean}$ and $2 SD < \text{mean}$. We then plotted the percent of sampled eBird checklists against the slope estimate for the continuous day model parameter (i.e., the trend slope estimate), to investigate whether or not these slope estimates converged.

This figure shows the first 10 species performed for this analysis (all top 50 species are shown in the supplementary material uploaded with the revised manuscript). The blue lines represent a quantile regression with 5 and 95% confidence limits. Note that the y-axis is different among species. It demonstrates that the slope estimates do converge, but at largely

different sample sizes (cf. Sulphur-crested Cockatoo versus Welcome Swallow). It also demonstrates that the range of slope estimates varies among species (i.e., the y-axis varies substantially among some species):

This figure shows the rank of species based on the maximum range in slope estimates – i.e., the ‘noise’ in slope estimates across many different models.

We are unaware of any statistical method which allows us to easily quantify when these slope estimates ‘converge’, and it is made more difficult because of the varying scale of the units of the slope among species. Hence, we further investigated this by making the y-axis ‘unitless’. To do this, we first calculated the ‘best estimate’ – e.g., the average slope estimate when using the maximum amount of data for each species, and then calculated the relative difference from that estimate as a function of sample size (i.e., percent of the possible checklists).

Because the y-axis means the same thing for every species now, the y-axis is fixed. This is the first 10 species, and all 50 species can be found in the supplementary material (Appendix 2). The blue lines again represent 5% and 95% confidence intervals. Clearly, the relative difference from the best estimate again varies among species, with some species not needing a large additional pool of eBird checklists to approach the best estimate (e.g., Pied Currawong) and other species needing more eBird checklists (e.g., Australian Raven). Visual inspection shows different critical thresholds for different species (cf., Welcome

Swallow and Rainbow Lorikeet). To quantify this, we calculated when the maximum absolute relative difference was reduced by at least 50% – a rough measure of convergence of the absolute difference – for each species, and calculated at what sampling level this occurred.

We found that the median number of checklists necessary for the convergence of slope estimates to 50% reduction in the absolute difference was ~11,700, with a min of ~ 6,500 and a max of ~ 20,800. This is roughly comparable with Horns et al. 2018 (i.e., ~ 10,000 eBird checklists). We note however, that these analyses are preliminary, but promising, and future work should investigate these questions further.

Although we do not delve into these results presented here in depth in the revised main manuscript (partly because of the strict page limit), we do provide the results in supporting information. We first highlight this analysis to the reader in the methods (lines 132 – 134):

“For the top 50 species in our analysis, we further investigated the robustness of these trend estimates in respect to sample size (Appendix 2).”

We then integrate these results into the discussion, highlighting the potential of identifying critical thresholds for a biodiversity model, and demonstrate that if these thresholds are known than effort could be directed based on these thresholds. This section reads (lines 286 – 295):

“Another critical component to efficiently direct effort and maximize the collective effort of citizen scientists is by understanding critical thresholds necessary for reliable estimates of trend detection. If the minimum number of sampling events for a region is understood, then citizen science effort could appropriately be directed to areas where these critical thresholds are not yet met. We preliminarily found that for the top 50 species in our analysis, ~ 11,700 checklists were necessary for a 50% reduction/convergence in the slope estimate based on our model (Appendix 2). This result is comparable with a study in the USA which found that ~ 10,000 eBird checklists were necessary to provide reliable trend estimates [31]. Future work should investigate critical thresholds for biodiversity analyses and how these interact with efficiently directing citizen science effort.”

Lastly, we see no reason that the number of observers would influence population trends as there is not necessarily a direct relationship between number of observers and number of eBird checklists. Most citizen science projects, including eBird (Wood et al. 2011), follow a “10% rule”, where 90% of the data comes from 10% of the contributors. This suggests that reliable trend inference is more likely to be a function of the number of checklists submitted than the number of observers.

- **REVIEWER 1:** “Rather than a criticism of their work, this is about being more circumspect about those taxa / regions best suited to their approach.”

Response: *We agree more work is needed to fully have a generalizable approach. Thus, we have toned down the generalizability of our results throughout the discussion, and thus believe we have been more circumspect about the taxa/regions that are best suited to this approach. For example, we removed the following sentence from the discussion (previously lines 220 – 222):*

“Our approach of relying on statistical leverage as a measure of the value of a citizen science observation can be used regardless of whether a citizen science project is global, regional, or carried out in the constraints of a local park.”

- **REVIEWER 1:** “Thus, rather than optimizing where data are collected within a particular region to detect trends, it may be that simply acquiring more data ANYWHERE in the region is the priority, attracting more effort from adjacent, better sampled areas.”

Response: *We interpret this comment as questioning the ‘spatial scale’ of our approach. In other words, if the analysis was completed on continental Australia, then regions with poorly sampled areas would possibly be more valuable and thus attract effort from adjacent better sampled areas. We agree that this is a strong possibility, but that more work will be necessary to answer this. For instance, if our analysis was conducted throughout all of Australia, it is likely that data from Longreach would be much more valuable than data from Brisbane or Sydney (refer to histogram figure above). However, we believe it probably isn’t practical to encourage sampling throughout large continental regions and our approach is more adaptable at a regional scale or within a large national park. We believe that this analysis is outside the current scope of our manuscript, but we have added this possibility in the manuscript (lines 247 – 259):*

“Nevertheless, our findings appear to be robust to spatial scale, at least within a regional level. It is critical to track biodiversity trends at multiple spatial scales [64], as biodiversity estimates sometimes change dependent on the spatial scale [14]. In comparison with other regions in Australia, the distribution of sampled grids in Sydney is generally similar – many grids unsampled or sampled only a few times, and then large variation among the rest of the grids (Appendix 5). Different regions have the same underlying ‘starting point’ in the current sampling regime, suggesting our results are generalizable among regions. Although this may only be applicable at a regional scale, and future work should further investigate these patterns at large continental-scales where the grain size is proportional to the spatial scale of the study. For example, within all of Australia, it is likely that unsampled regions will be significantly more important because there are many ‘gaps’ in the data, and effort could thus be directed from well sampled regions to unsampled regions.”.

- **REVIEWER 1:** it will also integrate their work with the extensive literature on sampling effort determination

Response: *We have expanded the scope of our work, and modified our wording to be more circumspect about the future of this exciting work with optimizing citizen science sampling for biodiversity. See detailed comments above for how we addressed these suggestions.*

REFERENCES

- Callaghan et al. 2017. <https://doi.org/10.5751/ACE-01104-120212>
Horns et al. 2018. <https://doi.org/10.1016/j.biocon.2018.02.027>
Bacaro et al. 2012. <https://doi.org/10.1016/j.ecocom.2012.05.007>

Referee: 2

Comments to the Author(s)

I believe that this is an important paper that addresses some crucial issues for citizen science, and does so in a clear way. I too am exploring these issues of making citizen science more 'adaptive' by incentivising recorders to make visits at the times and places where their records will be more informative. This paper is a good contribution to this and (I think) the first published example. I like the fact that this paper simplifies this potential complex analysis to some 'rules of thumb' about what is most important and makes some specific recommendations.

Response: *We appreciate this positive and thoughtful critique of our work.*

The Introduction is very well-written.

Methods:

I think it would be helpful in the Methods to have an overview of what you did. I had to re-read the Methods several times to fully understand exactly what you did. I think you: 1. took 25995 checklists and calculated linear trends for 235 species; 2. calculated the statistical leverage for the 25995 x 235 combinations of species observations; 3. summed the leverages for each checklist to give an overall importance for each checklist; 4. related the importance of the checklists to their traits; 5. predicted the expected leverage on a daily basis for grid cells in the region.

Response: *The reviewer is correct in their interpretation of the methods. We agree with this suggestion and decided to put this methodological overview in the last paragraph of the introduction, explicitly highlighting to the reader the steps that are further laid out in the methods. This section now reads (lines 91 – 99):*

“To do this, we: (1) used 25,995 eBird citizen science sampling events; (2) analyzed linear trends for 235 species; (3) calculated a measure of statistical leverage (i.e., marginal value) – the influence a given observation has on the population trend model of a species – for all checklists for each species; (4) summed the leverages on a given checklist to provide a measure of marginal value for every checklist (i.e., the cumulative value/importance of a sampling event to inform our total knowledge of species’ trends, across many species); (5) tested specific predictions (Appendix 1) which may influence the marginal value of a citizen science sampling event [12]; and (6) used these associations to predict the expected marginal value on a daily basis.”.

L125 please note that you modelled a linear trend with your logistic regression. This does not account for seasonality in species (or their detection). It is not suitable for long-term trends which are unlikely to be linear. This should be addressed in the Discussion because your framework could be applied to other model outputs.

Response: *The reviewer is correct that our framework could be applied to other model outputs. We have adopted this suggestion, and added the following to the methods (lines 129 – 131):*

“This specific linear model may not be suitable for species which have varying detection probabilities throughout the full-annual cycle, but a large suite of models is possible in our framework.”.

And also added the following to the discussion, where we discuss our framework’s generalizability (lines 303 – 308):

“But these different outcomes can still be quantified in the same framework we introduce here: this framework could be applied to a wide suite of statistical models – including for different taxa and including more complicated trend analysis accounting for intra-annual varying detection probabilities. The key piece of information is some form of statistical leverage that can be calculated from a potential statistical model.”.

L133 Please rephrase "the probability of observation of that species with respect to time" for clarity.

Response: *We have rephrased this section. It now reads (lines 141 – 142):*

“Because for trend detection, we are interested in one specific model parameter—the temporal component—dfBeta rather than Cook’s Distance is appropriate [8].”.

L140-143 suggest change "given" to "each" for clarity.

Response: *We have rephrased this sentence for clarity. It now reads (lines 152 – 153):*

“In our case, every sampling event for each species received a dfBeta value (i.e., each species received 25,995 measures of dfBeta), using the ‘dfBetas’ function from R [57].”.

L164 These correlations should be given in an Appendix.

Response: *We have now included them in the supporting information (Appendix 3). We also include them here, for the reviewer’s reference.*

5 km grid size

10 km grid size

25 km grid size

50 km grid size

I suggest that you move the hypotheses into the main text (if you have space), ideally with the summary statistics for the parameters.

Response: *We initially had these predictions in the main text, but unfortunately these made the most sense to 'cut' to restrict our page limit to 10 pages as the journal requires. After adding more depth in other sections (e.g., in response to Reviewer 1's comments), we are still 'short-on-space'. Hence, we have left these predictions in the Appendix, but we do feel this is better integrated into the general overview of the methods that has now been provided (see comments above). We do find it necessary to remind the reader that the predictions are easily read in the Appendix, and thus we reference Appendix 1 three times now and added the following explicit clause in the discussion (lines 218 – 220):*

“For those sampled sites, however, we found a generally positive relationship between a number of our predicted parameters (detailed predictions can be found in Appendix 1) and the relative value of a sampling event (Fig. 2).”

See comment below (one starting with “L210”) for elaboration on the summary statistics for the predictors/parameters.

Results and Discussion

You brought out some simple messages from this analysis nicely in the Discussion, e.g. the value of sampling new locations v revisiting locations.

Response: *Thank you for this positive feedback. We believe that this work is the beginning of some potentially very useful questions for the future of citizen science sampling.*

L193 Really interesting thought. Is there circularity here: revisited sites are valuable because they provide much information, hence they appear to be hotspots, hence recorders are "already sampling the 'hotspots'"?

Response: *This is a good question, that is potentially very difficult to answer. We believe that naturalists, in general, are 'good at finding things', so-to-speak. In other words, a birder knows where to go find birds, and they will generally avoid the parts of their region that are devoid of birds. We believe that this effect is what was driving the patterns that we highlighted in this section. But, it is possible that there is a 'popularity' effect whereby some spots are revisited because they have been hotspots in the past. In the birding world, this is sometimes referred to as the Patagonia Picnic Table Effect*

(https://en.wikipedia.org/wiki/Patagonia_picnic_table_effect#targetText=The%20Patagonia%20picnic%20table%20effect,better%20than%20other%20similar%20localities.). We have further qualified this sentence though to highlight the potential circularity in this argument. The section now reads (lines 212 – 216):

“In other words, citizen scientists are likely already sufficiently sampling biodiversity in space: they appropriately identify and sample 'hotspots' in space that should receive the most sampling attention. But the effect of citizen scientists visiting

‘popular’ locations (e.g., spots known for their bird diversity) could exclude the discovery of other known ‘hotspots’ in the same region.”.

L187-190 Can you please include the effect sizes. Actually, I wonder if visiting new sites becomes increasingly important as the grain size increases? (i.e. visiting a new 5km is not valuable, visiting a new 25km square is valuable.)

Response: *We have added effect sizes, and the Reviewer was correct in their prediction. We have also added this into the sentence. The section now reads (lines 204 – 209):*

“We found weak evidence that visiting an unsampled site was marginally more valuable than visiting an already sampled site, but we did find that as grain size increases the importance of sampling unsampled sites also increases. There was no statistical clarity for the 5 km ($p=0.669$; effect size = -0.25 ± 0.58), and 10 km ($p=0.093$; effect size = 1.27 ± 0.76) grain sizes, but there was for the 25 km ($p=0.035$; effect size = 2.98 ± 1.42) grain size. At the 50 km grain size, this test was not possible because all sites had been sampled.”.

L197ff You have done a good job of summarising the results, but I think you should acknowledge that there is quite a lot of variation in these predictors (cp L217 "generally consistent"). No predictor is consistently significant across all grain sizes.

Response: *We agree with this, and in general are more circumspect about our results throughout the discussion (see detailed reply to Reviewer 1 above). In this particular section, we have reworded and acknowledged the variability that exists as well as elaborated on the potential robustness of spatial scale in our analysis (lines 246 – 259):*

“We found generally consistent results, albeit with variation in parameters: no predictor was consistently significant across all grain sizes. Nevertheless, our findings appear to be robust to spatial scale, at least within a regional level. It is critical to track biodiversity trends at multiple spatial scales [64], as biodiversity estimates sometimes change dependent on the spatial scale [14]. In comparison with other regions in Australia, the distribution of sampled grids in Sydney is generally similar – many grids unsampled or sampled only a few times, and then large variation among the rest of the grids (Appendix 5). Different regions have the same underlying ‘starting point’ in the current sampling regime, suggesting our results are generalizable among regions. Although this may only be applicable at a regional scale, and future work should further investigate these patterns at large continental-scales where the grain size is proportional to the spatial scale of the study. For example, within all of Australia, it is likely that unsampled regions will be significantly more important because there are many ‘gaps’ in the data, and effort could thus be directed from well sampled regions to unsampled regions.”.

L210 Please provide some summary statistics about each of the predictors (e.g. range, inter-quartile range and median)

Response: We have provided the summary statistics for each of the 5 predictors in the main manuscript, and have also provided a more full-picture of the summary statistics in the supporting information for the manuscript (Appendix 4). The relevant section in the manuscript now reads (lines 220 – 234):

“Full summary statistics for each of our predictors can be found in Appendix 4, but the range, median, and interquartile range, respectively, can be found in parentheses after the referenced parameter. The number of unique days sampled (5km: 1—1222, 25, 228; 10km: 1—1894, 79, 424; 25km: 1—2946, 1103, 2071.5; 50km: 11—3004, 2284, 808) — likely represented from known ‘hotspots’ identified by citizen scientists — had the strongest, positive, effect size, and this was robust to grain size comparisons. The median sampling interval (5km: 1—2450, 31, 124; 10km: 1—1401, 11, 57; 25km: 1—821, 1, 5; 50km: 1—193, 1, 0) was also strongly associated with high value samples, with an exception at the 25 km grain size. Distance to the nearest sampled site (5km: 1.9—19.7, 5, 2.1; 10km: 0—23.1, 10, 0.7; 25km: 14.8—25, 22, 4.3; 50km: 32.3—45.1, 37.5, 3.6) and the nearest-neighbor sampling interval (5km: 1—2450, 43.5, 170; 10km: 1—1401, 14, 54; 25km: 1—821, 2, 5; 50km: 1—1, 1, 0) influenced the value of a sampling event less than the other parameters. Surprisingly, the number of days since the last sample (5km: 1—2935, 39, 228; 10km: 1—1645, 10, 86; 25km: 1—693, 1, 4; 50km: 1—316, 1, 0), while positively associated, had less influence than other parameters.”

L212 Interestingly, this recommendation (to 'left shift' the distribution) is a form of structuring.

Response: We agree that it would be a form of structuring, but in a dynamic way, as opposed to how most citizen science projects generally view ‘structure’ currently. We have added an additional clause to this sentence, reflecting this (lines 242 – 244):

“A secondary goal should be to decrease (i.e., left-shift the distribution) the median sampling interval of sites; providing some structure to unstructured citizen science projects.”

L213ff If the inflation effect of the history of the site is true, then I'd expect this to be consistent. But the effect of number of days sampled is not consistent at the next largest grain size (25km). Can you explain this discrepancy?

Response: This is a good point, and it appears that the 25km grain size was a bit of an ‘outlier’ in our analysis. It is possible that the inflation effect may not be true, but it is obvious that the number of unique days sampled is a critical component across all grain sizes (significant for three out of four grain sizes). As such, we have removed this sentence from the manuscript and simply left our conclusion that the ‘history’ of a site is generally important across grain sizes. We added a clarifier in the previous sentence highlighting the importance of the “history” of a site (lines 239 – 242):

“The ‘history’ of a site is particularly important while considering whether to sample that site: the number of unique days sampled was the strongest predictor for all but the 25 km (second strongest) grain size, suggesting that observations from sites with a long time-series are relatively more valuable.”

L217 Pedantically, this is not the size of the 'site', but the grid size used for aggregation and analysis.

Response: *We agree. This has now been reworded based on a previous comment, and thus is no longer applicable. Further, we adopted your suggestion of ‘grain size’ throughout the manuscript when referring to our results, except when ‘site’ makes more sense such as the applications of our analysis.*

L234 "look backward first". Please clarify this - it was not immediately obvious what you meant. I think you mean "firstly analysed past data... in order to make predictions during the following year",

Response: *You are correct in your interpretation. We have rephrased this section. This now reads (lines 269 – 271):*

“This approach required us to analyze data from the past first, using a model with all observations for 2018, based on statistical leverage calculated from 2010-2018, in order to predict the expected marginal value for any given day.”

L242 and Fig 3. You use 'expected values' - but of what? - I suggest selecting a more descriptive term.

Response: *We have rephrased this to “expected marginal value” as we chose to use marginal value throughout the manuscript, which is the same as statistical leverage. This sentence now reads (lines 276 – 277):*

“Instead of participants preferentially chasing specific species, this approach would guide participants to the sites with the highest expected marginal value for the biodiversity dataset.”

Fig 2. Please expand the legend so that it is fully understandable on its own.

Response: *We have expanded this figure legend so that it is understandable on its own. It now reads:*

“Figure 2. The parameter estimates (and 95% confidence intervals) for four separate linear models (i.e., at each of the respective grain sizes), showing the relative strength of the results and effect sizes for each of the predictors. Full summary statistics for each of the predictors can be found in Appendix 4. Variables were log-transformed and then standardized, allowing for direct comparison among effect sizes.”

Fig S1. This is a valuable figure that could be brought into the main text, if there is space. What is 'marginal value'? You mention it in L146, but I think it is a synonym for 'leverage' and 'expected value'? I prefer using one term for one idea.

Response: *We originally had this figure in the main manuscript but was cut for space restrictions, and thus agree with the Reviewer. As such, we have brought it back into the main manuscript as an additional panel with Figure 3, since they are showing more-or-less the same thing just one spatially and one temporally. We have rephrased the manuscript throughout to be more clear with our use of leverage/value. See previous comment regarding this.*

I thought that the Discussion was strong in terms of the practical recommendations and next steps.

Response: *We appreciate this positive feedback, and thank the reviewer for their detailed critique which has improved our manuscript.*

Appendix B

Professor Spencer Barret, FRS
Department of Ecology and Evolutionary Biology
University of Toronto
Editor, The Royal Society Proceedings B

September 8th, 2019

Dear Professor Barret,

Thank you for this great news about our manuscript. We are very excited to have it published in Proceedings B. We have responded to the Associate Editor's minor comments, below.

Sincerely,
Corey Callaghan

Centre for Ecosystem Science
School of Biological, Earth and Environmental Sciences
UNSW Sydney
E: c.callaghan@unsw.edu.au
P: +61 421 601 388

Associate Editor:

Board Member

Comments to Author:

I was impressed with the thorough measures taken by the authors to implement the recommendations of the two referees, and agree that the additional supplementary material provides valuable context for the results. I have two minor recommendations - one concerning format, and the other concerning clarification of the effects of one variable.

1. References should be numbered in citation order rather than alphabetical order.

Response: *We have now redone the reference order for the final submission.*

2. Lines 220-237: please check that the units and directions of effects are clear in these summary statistics. Within this section (and elsewhere), the main effect I would like to see clarified is that of median sampling interval (also referred to as median waiting time in Appendix 3). In Appendix 1 you hypothesise that, “the median sampling interval would be positively associated with the value of a citizen science observation; i.e., observations from sites with high median waiting times would be more valuable” but from your results I understand that the median sampling interval had a negative (shorter intervals, stronger value) effect, because you later state, “A secondary goal should be to decrease (i.e., left-shift the distribution) the median sampling interval of sites; providing some structure to unstructured citizen science projects.” (lines 242-243). I might however have misunderstood this point, as your conclusion states “largest median sampling intervals both positively correlate with the relative marginal value” (lines 321-322) – so please make sure the descriptions of this effect are clear and unambiguous.

Response: *The summary statistics (lines 220-237) are correct, and we added a sentence to remind the reader that the effect sizes are in figure 2. This says “See figure 2 for standardized parameter estimates (i.e., effect sizes).” In line 235. The point in lines 242-243 was admittedly confusing. We have modified this sentence to be clearer, alleviating the potential for confusion: “Because sites with larger median sampling intervals were positively associated with the marginal value of a citizen science observation, a secondary goal could be to decrease (i.e., left-shift the distribution) the median sampling interval of sites by targeting sites with the largest median sampling intervals; providing some structure to unstructured citizen science projects.” We left lines 321-322 as-is, confirming the previous sentiment, which should now be more clear.*